# Short Chains, Deep Thoughts: Balancing Reasoning Efficiency and Intra-Segment Capability via Split-Merge Optimization

**Runquan Gui** [1 2]    **Jie Wang** [1] [†]    **Zhihai Wang** [3]    **Chi Ma** [1]    **Jianye Hao** [4]    **Feng Wu** [1]

## Abstract

While Large Reasoning Models (LRMs) have demonstrated impressive capabilities in solving complex tasks through the generation of long reasoning chains, this reliance on verbose generation results in significant latency and computational overhead. To address these challenges, we propose **CoSMo** (**Co**nsistency-Guided **S**plit-**M**erge **O**ptimization), a framework designed to eliminate structural redundancy rather than indiscriminately restricting token volume. Specifically, CoSMo utilizes a split-merge algorithm that dynamically refines reasoning chains by merging redundant segments and splitting logical gaps to ensure coherence. We then employ structure-aligned reinforcement learning with a novel segment-level budget to supervise the model in maintaining efficient reasoning structures throughout training. Extensive experiments across multiple benchmarks and backbones demonstrate that CoSMo achieves superior performance, improving accuracy by **3.3** points while reducing segment usage by **28.7%** on average compared to reasoning efficiency baselines.

## 1. Introduction

Recent advancements in Large Reasoning Models (LRMs) have advanced capabilities across domains such as mathematical reasoning, code generation, and complex planning (Guo et al., 2025; Team et al., 2023; Jaech et al., 2024). However, the extension of reasoning chains introduces substantial computational overhead, while the accumulation of redundant information frequently impedes reasoning capabilities. These issues are particularly acute in knowledge-intensive multi-hop question answering (QA). Such susceptibility arises because the task is highly sensitive to extraneous entities and redundant reasoning steps, which can disrupt the coherence of multi-hop inference (Kumar et al., 2024; Sui et al., 2025). Empirical evidence indicates a significant decline in accuracy as the number of reasoning segments increases beyond the ground-truth hop count (Yadav et al., 2025). Therefore, the primary challenge shifts from scaling the quantity of reasoning units to identifying an efficient reasoning paradigm that reduces redundancy while enhancing model performance.

Diverse strategies aim to optimize the reasoning efficiency and accuracy of LRMs within multi-hop QA. Pruning-based methods generally operate by refining fully generated reasoning chains, where statistical metrics or external LLM judges are employed to excise redundant components before fine-tuning the model on these condensed trajectories (Li et al., 2025; Munkhbat et al., 2025). However, such rigid reductionist approaches risk disrupting the logical coherence of the reasoning process (Cui et al., 2025). Reinforcement learning methods offer another path by integrating length constraints directly into the optimization objective, utilizing soft (Aggarwal & Welleck, 2025) or hard (Hou et al., 2025) penalties to encourage brevity. We posit that direct length penalization is suboptimal. Because deep reasoning inherently requires a greater volume of tokens, indiscriminate penalization of length can inadvertently suppress the emergence of complex reasoning capabilities essential for solving intricate problems.

To address these challenges, we propose **CoSMo** (**Co**nsistency-Guided **S**plit-**M**erge **O**ptimization), a framework that enforces reasoning consistency via dynamic structural refinement. Drawing inspiration from the classic split-and-merge algorithm in image segmentation (Horowitz & Pavlidis, 1976), CoSMo treats the reasoning chain as a sequence of discrete reasoning segments. The framework dynamically optimizes the structure by merging redundant segments and splitting logical gaps into coherent sequences. Unlike rigid pruning strategies that merely filter steps, this algorithm iteratively refines the reasoning topology until the process ensures both conciseness and logical continuity. Following data curation and supervised fine-tuning

[1]University of Science and Technology of China [2]<rqgui@mail.ustc.edu.cn> [3]Alibaba Group [4]College of Intelligence and Computing, Tianjin University. Correspondence to: Jie Wang <jiewangx@ustc.edu.cn>.

*Proceedings of the 43rd International Conference on Machine Learning*, Seoul, South Korea. PMLR 306, 2026. Copyright 2026 by the author(s).

(SFT), we optimize the model using group relative policy optimization (GRPO) with a novel segment-level budget. This mechanism specifically penalizes the quantity of redundant segments while leaving the token length within each segment unconstrained, thereby accommodating the token consumption essential for deep reasoning.

Experimental results substantiate the effectiveness of CoSMo. Specifically, our method reduces average token consumption by 29.5% while boosting accuracy by 2.8 points across HotpotQA (Yang et al., 2018), HaluEval (Li et al., 2023), NQ (Kwiatkowski et al., 2019), and CRAG (Yang et al., 2024). Compared to other efficient reasoning methods, CoSMo achieves a 28.7% reduction in average segments, demonstrating its unique efficacy in eliminating structural redundancy. Moreover, ablation studies reveal that the model fine-tuned solely via the split-merge algorithm attains the lowest reasoning redundancy and highest accuracy compared to other SFT baselines. Subsequent reinforcement learning further enhances intrinsic model capabilities. Crucially, on OOD datasets with increasing difficulty, CoSMo exhibits exceptional robustness as complexity rises, highlighting the promising potential of the low-redundancy reasoning paradigm for tackling harder tasks.

The core contributions are summarized as follows:

- **Consistency-Guided Optimization**: We propose CoSMo, a framework employing a split-merge algorithm to ensure reasoning conciseness while strictly preserving logical coherence. By merging redundant segments and decomposing logical leaps under consistency guidance, CoSMo dynamically refines the reasoning topology to align with intrinsic problem complexity.

- **Segment-Level Budgeting**: We introduce a segment-level penalty mechanism within reinforcement learning to replace token-length constraints. This design penalizes structural redundancy while permitting the model to preserve the descriptive capacity necessary for deep reasoning during training.

- **Empirical Effectiveness**: We empirically demonstrate the effectiveness of CoSMo across multiple backbones. The framework achieves an average accuracy improvement of **+3.3** points across four benchmarks and a **28.7%** reduction in structural redundancy relative to existing efficient reasoning baselines.

## 2. Related Work

**Reasoning Paradigms** To enhance reasoning capabilities, various paradigms extend the linear chain-of-thought (CoT) approach. Methods such as ToT (Yao et al., 2023), GoT (Besta et al., 2024), and HTP (Gui et al., 2025) intro-

duce complex structures like trees and graphs to broaden search spaces. While effective for general LLMs, the utility of such structural interventions diminishes for LRMs that inherently possess robust reasoning patterns. Recent research shifts toward regulating the reasoning process through conciseness constraints (Renze & Guven, 2024; Xu et al., 2025; Ding et al., 2024) or budget controls (Han et al., 2025). However, these prompt-based strategies often suffer from coarse granularity and fail to precisely balance conciseness with correctness, frequently resulting in performance degradation due to over-simplification.

**Fine-Tuning for LRMs** Current fine-tuning paradigms for LRMs primarily center on the construction and curation of high-quality reasoning data. For instance, Logical DA (Zheng et al., 2025) employs a multi-agent framework to synthesize logically rigorous samples, while RATIONA-LYST (Jiang et al., 2025) trains auxiliary models to explicate implicit logical steps. Filtering metrics based on reasoning length (Shen, 2024) and perplexity (Ankner et al., 2024) are utilized to curate high-value instances. Recent research attention, however, has increasingly pivoted toward optimizing inference efficiency alongside correctness (Feng et al., 2025; Liu et al., 2025a). Similarly, several approaches distill concise rationales by leveraging stronger teacher models to eliminate redundancy (Xia et al., 2025; Kang et al., 2025), or by self-training on the shortest successful trajectories sampled from the model itself (Liu et al., 2024; Munkhbat et al., 2025; Ma et al., 2025; Jin et al., 2025). Statistical heuristics further aid in this compression, pruning steps characterized by low entropy (Li et al., 2025) or high perplexity (Cui et al., 2025) to reduce redundancy.

Beyond supervised methods, reinforcement learning techniques directly optimize generation length, employing either soft length-penalties in the reward function (Aggarwal & Welleck, 2025; Hou et al., 2025; Shen et al., 2025) or hard constraints that assign zero rewards for budget violations (Hou et al., 2025). Nevertheless, we argue that imposing direct length penalties within reinforcement learning is fundamentally suboptimal. Since valid reasoning segments often demand elaboration to ensure accuracy, indiscriminate length penalties punish necessary details, thereby stifling the development of deep reasoning capabilities. In contrast to (Gui et al., 2026), which enhances trustworthiness through step-wise semantic verification at the expense of additional computational overhead, CoSMo is dedicated to achieving efficient reasoning by optimizing structural redundancy.

## 3. Preliminary

In this section, we first formalize the problem of knowledge-intensive multi-hop QA in Section 3.1, establishing a hierarchical definition of reasoning segments. Subsequently, in

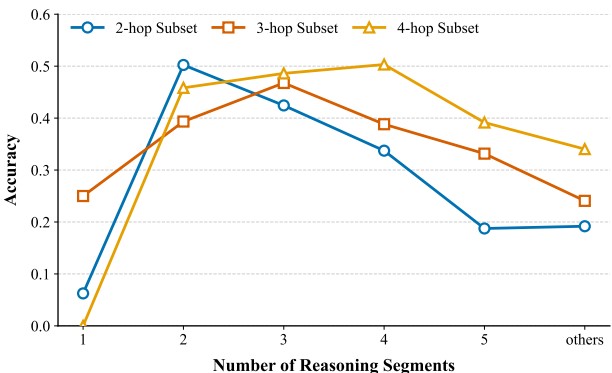

*Figure 1.* Impact of enforced reasoning segment counts on accuracy across MuSiQue subsets.

Section 3.2, we present empirical findings that highlight the critical impact of reasoning depth on model performance, serving as the motivation for our proposed framework.

### 3.1. Problem Statement

**Task Definition.** We anchor our study in the domain of multi-hop QA, where queries intrinsically require multi-step deduction and annotated reasoning paths are available. Let $\mathcal{D} = \{(q, R^*, a^*)\}$ denote a dataset, where $q$ represents the input query and $a^*$ is the correct answer. $R^*$ represents the reference ground-truth reasoning trajectory. We define the requisite ground-truth hop count as $k^* = |R^*|$, representing the number of discrete reasoning steps, which serves as the gold standard for structural complexity. Formally, the model $\pi_\theta$ takes the query $q$ as input and generates a predicted reasoning chain $R$, concluding with a final answer $\hat{a}$.

**Hierarchical Reasoning Structure.** Unlike standard formulations that treat the reasoning chain $R$ strictly as a flat sequence of tokens, we model it as a sequence of discrete reasoning segments. Let $R = (s_1, s_2, \ldots, s_N)$ denote a chain of $N$ generated segments. Each segment $s_i$ is composed of a variable sequence of tokens $s_i = (w_{i,1}, \ldots, w_{i,L_i})$, where $L_i$ represents the token length of the $i$-th segment. The generation probability of $R$ is decomposed at the segment level: $P_\theta(R \mid q) = \prod_{i=1}^{N} P_\theta(s_i \mid q, s_{<i})$, where $s_{<i}$ denotes the history of preceding segments. This formulation explicitly decouples *structural redundancy*, represented by the segment count $N$, from *intra-segment reasoning*, represented by the descriptive granularity $L_i$. This distinction is pivotal as it enables the optimization of reasoning depth without constraining the descriptive capacity required for individual segments.

### 3.2. Preliminary Findings: The Impact of Reasoning Topology

To empirically validate the relationship between reasoning topology and task performance, we conducted a controlled

experiment on the MuSiQue (Trivedi et al., 2022) benchmark, utilizing its stratified subsets of 2-hop, 3-hop, and 4-hop questions. We employed Llama-3.1-8B-Instruct (Dubey et al., 2024) as the backbone model to generate reasoning chains under standard prompting conditions. Subsequently, to decouple the impact of reasoning structure from model capability, we stratified the generated responses into six distinct cohorts based on the number of reasoning segments, specifically ranging from 1 to 5 segments, along with an aggregate category for longer chains.

As illustrated in Figure 1, the results exhibit a distinct consistency between model capability and the reasoning structure. Specifically, the model achieves global maximum accuracy when the segment count aligns with the ground-truth hop count, while performance degrades rapidly upon the introduction of redundant segments. For instance, regarding 2-hop questions, accuracy peaks at 50.23% when $N = 2$ but declines sharply to 42.44% when the chain is extended to 3 segments. This observation empirically underscores that efficiency in knowledge-intensive QA is not monotonically related to brevity. Instead, there exists an intrinsic structural optimum. Deviating from this optimum compromises reasoning robustness, regardless of whether such deviation results from compression or unnecessary expansion. These findings motivate our CoSMo framework, which seeks to automate this structural alignment. Detailed experimental setups are provided in Appendix A.4.

## 4. Methodology

**Overview of CoSMo.** In this section, we introduce CoSMo in detail. We first establish the theoretical foundation by decoupling structural complexity from intra-segment reasoning in Section 4.1. Guided by this principle, Section 4.2 introduces the split-merge optimization algorithm to curate logic-dense data for SFT. Finally, Section 4.3 details the online structure-aligned reinforcement learning process using GRPO to robustly align the model's reasoning topology.

### 4.1. Decoupling Structure from Intra-Segment Reasoning

Existing strategies for efficient reasoning, irrespective of their specific implementation, rest upon a common theoretical premise. They treat the reasoning chain $R$ as a homogeneous sequence of tokens, effectively conflating efficiency with the minimization of cumulative token volume. Formally, let $R$ be the reasoning chain generated by policy $\pi$ for a query $q$, where $L_i$ denotes the token length of the $i$-th segment. The generalized optimization objective can be abstracted as finding a policy $\pi$ that satisfies:

$$\min_{\pi} \; \mathbb{E}_{q \sim \mathcal{D}} \left[ \mathbb{E}_{R \sim \pi(\cdot|q)} \left[ \sum_{i=1}^{N} L_i \right] \right] \quad \text{s.t. } \mathcal{A}(\pi) \geq \mathcal{A}_0 \quad (1)$$

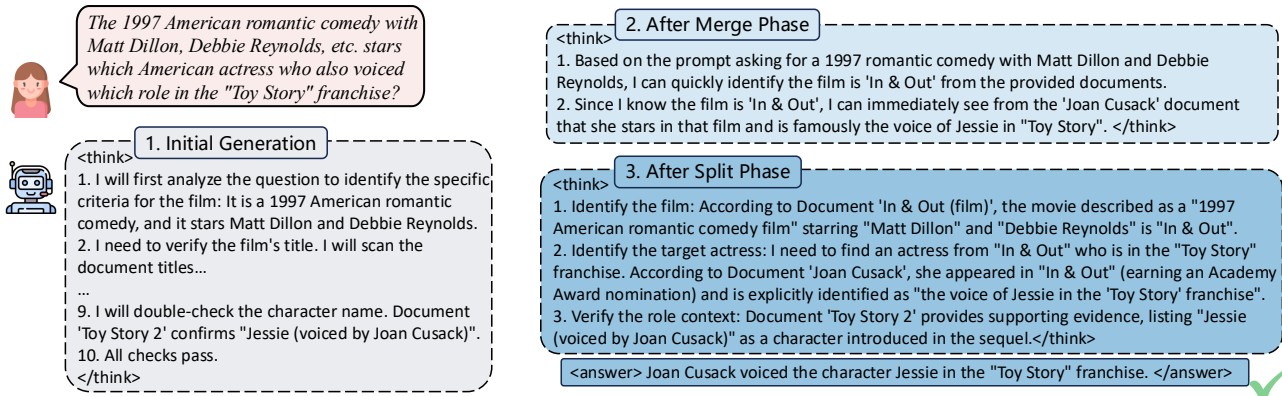

*Figure 2.* The process of Merge and Split phases (see Appendix D for the full case study).

where $\mathcal{A}(\pi)$ represents the accuracy on dataset $\mathcal{D}$, and $\mathcal{A}_0$ denotes the baseline performance. Under this objective, optimization pressure applies indiscriminately to the total length. The model is compelled to compress the reasoning process across two dimensions simultaneously: structural depth ($N$) and intra-segment reasoning ($L_i$). This often results in a suboptimal trade-off, where the model sacrifices the descriptive detail required for complex logical transitions to adhere to the global token budget.

In contrast, guided by the findings in Section 3.2, we define reasoning efficiency strictly by topology rather than token granularity. We decouple optimization from cumulative length, focusing instead on aligning the segment count $N$ with the ground-truth complexity $k^*$:

$$\min_{\pi} \mathbb{E}_{q \sim \mathcal{D}} \left[ \mathbb{E}_{R \sim \pi(\cdot|q)} \left[ |N - k^*| \right] \right] \quad \text{s.t.} \quad \mathcal{A}(\pi) \geq \mathcal{A}_0 \quad (2)$$

By excluding segment length $L_i$ from the objective, this formulation targets structural redundancy (driving $N \to k^*$) while leaving intra-segment reasoning unconstrained. This grants the model flexibility to reasoning on details necessary for validity without structural penalties.

### 4.2. Split-Merge Optimization Algorithm

The core mechanism of CoSMo involves automatically curating a high-quality, logic-dense dataset, $\mathcal{D}_{\text{cosmo}}$, from raw model outputs. We propose an iterative optimization algorithm that dynamically reshapes the reasoning chain to converge toward the gold logical depth $k^*$. The algorithm unfolds across three sequential stages: *structured generation*, *iterative optimization*, and *fine-tuning*. This overall process is illustrated in Figure 2.

**(1) Structured generation and filtering**. The process commences with the acquisition of high-quality reasoning seeds. Given the input query $q$, we prompt the backbone model $\pi_\theta$ to generate reasoning chains formatted as numbered lists (e.g., "1.", "2."), yielding inherently discretized chains

$R_{\text{init}} = (s_1, s_2, \ldots, s_M)$. From these generated candidates, we enforce strict filtering for correctness, retaining only those trajectories where the predicted answer $\hat{a}$ matches the ground truth $a^*$ exactly. These validated seeds serve as the foundation for our subsequent optimization phase.

**(2) Iterative split-merge optimization**. This phase constitutes the core mechanism of CoSMo. Modeling the reasoning chain $R$ as a dynamic sequence, we iteratively apply merge and split operators to align the segment count $N$ with the gold hop count $k^*$. To ensure semantic fidelity and logical soundness, we employ two distinct LLM-based modules: the consistency judge $\mathcal{M}_{\text{judge}}$, which outputs a binary verdict $v \in \{0, 1\}$ regarding logical topology, and the semantic generator $\mathcal{M}_{\text{gen}}$ for rewriting content. The optimization process is guided by the structural deviation $\Delta = N - k^*$.

*The Merge Phase (Semantic Fusion):* When the chain exhibits structural redundancy ($N > k^*$), we seek to consolidate segments without information loss. We traverse adjacent segments $(s_i, s_{i+1})$ and prompt $\mathcal{M}_{\text{judge}}(s_i, s_{i+1})$ to assess their relationship. Rather than using rigid thresholds, the judge directly identifies semantic redundancy, specifically determining whether $s_{i+1}$ is merely a paraphrase or trivial extension of $s_i$. If redundancy is confirmed, semantic fusion is triggered. Unlike simple concatenation, the generator synthesizes a unified logical unit:

$$s_{\text{new}} = \mathcal{M}_{\text{gen}}(s_i, s_{i+1}, \texttt{<fuse>})$$

This integrates the pair's semantic content into a single concise segment $s_{\text{new}}$, reducing the segment count by 1.

*The Split Phase (Logical Decomposition):* In contrast, an overly compressed chain ($N < k^*$) implies the presence of coarse-grained logic. We examine each segment $s_i$ using $\mathcal{M}_{\text{judge}}(s_i)$ to assess its reasoning granularity. If the judge detects multiple implicit jumps or composite reasoning within $s_i$, logical decomposition is triggered. The generator explicitly unpacks the compressed logic into a sequence

**Algorithm 1** Consistency-Guided Split-Merge Optimization

---

**Input** : Query $q$, Gold Hops $k^*$, Initial Chain $R = \{s_1, \dots, s_n\}$, Judge $\mathcal{M}_{\text{judge}}$, Generator $\mathcal{M}_{\text{gen}}$, MaxIter $T_{\max}$
**Output** : Refined Reasoning Chain $R_{\text{refined}}$
$t \leftarrow 0$; $N \leftarrow |R|$; $modified \leftarrow$ True
**while** $t < T_{\max}$ **and** $N \neq k^*$ **and** $modified$ **do**
$\quad$ $modified \leftarrow$ False
$\quad$ **if** $N > k^*$ **then** $\qquad$ // Merge Phase
$\quad\quad$ **for** $i \leftarrow 1$ **to** $N-1$ **do**
$\quad\quad\quad$ **if** $\mathcal{M}_{judge}(s_i, s_{i+1})$ *detects redundancy* **then**
$\quad\quad\quad\quad$ $s_{\text{new}} \leftarrow \mathcal{M}_{\text{gen}}(s_i, s_{i+1}, \texttt{<fuse>})$
$\quad\quad\quad\quad$ $R \leftarrow \text{Replace}(R, (s_i, s_{i+1}), s_{\text{new}})$
$\quad\quad\quad\quad$ $modified \leftarrow$ True
$\quad\quad\quad\quad$ **break** $\qquad$ // Re-evaluate
$\quad\quad\quad$ **end**
$\quad\quad$ **end**
$\quad$ **else if** $N < k^*$ **then** $\qquad$ // Split Phase
$\quad\quad$ **for** $i \leftarrow 1$ **to** $N$ **do**
$\quad\quad\quad$ **if** $\mathcal{M}_{judge}(s_i)$ *detects coarse logic* **then**
$\quad\quad\quad\quad$ $S' \leftarrow \mathcal{M}_{\text{gen}}(s_i, \texttt{<expand>})$
$\quad\quad\quad\quad$ $R \leftarrow \text{Replace}(R, s_i, S')$
$\quad\quad\quad\quad$ $modified \leftarrow$ True
$\quad\quad\quad\quad$ **break** $\qquad$ // Re-evaluate
$\quad\quad\quad$ **end**
$\quad\quad$ **end**
$\quad$ $N \leftarrow |R|$
$\quad$ $t \leftarrow t + 1$
**end**
**return** $R_{\text{refined}} \leftarrow R$

---

of finer-grained segments:

$$(s'_{i,1}, s'_{i,2}, \dots) = \mathcal{M}_{\text{gen}}(s_i, \texttt{<expand>})$$

This decomposes the coarse segment $s_i$ into a coherent sub-chain, fully articulating the intermediate reasoning and increasing the segment count.

*Iterative Refinement:* These operators are applied iteratively. In each iteration, the algorithm re-evaluates the segment count $N$ against the target $k^*$. The process concludes when the chain reaches the target depth or a semantic local optimum where no valid operations remain. This results in a refined chain $R_{\text{refined}}$ that is structurally aligned with the ground truth and semantically optimized for clarity. The detailed procedure is presented in Algorithm 1.

**(3) Supervised Fine-Tuning**. Upon completion of the iterative refinement, we consolidate the curated samples into the dataset $\mathcal{D}_{\text{cosmo}} = \{(q, R_{\text{refined}}, a^*)\}$. We then fine-tune the backbone model $\pi_\theta$ on this dataset using standard maximum likelihood estimation to obtain the supervised policy $\pi_{\text{SFT}}$. This phase is critical for instilling a structural prior, enabling the model to produce reasoning chains that naturally approx-

imate the optimal logical depth. This supervised training establishes a robust policy initialization for the subsequent reinforcement learning phase.

### 4.3. Structure-Aligned Reinforcement Learning

Following supervised fine-tuning, the model $\pi_{\text{SFT}}$ has established a foundational structural prior. To further enhance model capabilities, we employ Group Relative Policy Optimization (GRPO) (Guo et al., 2025). This efficient online algorithm optimizes the policy through group-wise reward normalization, thereby obviating the need for a separate value function critic.

Given a query $q$, we initialize the policy $\pi_\theta$ with $\pi_{\text{SFT}}$. During training, we sample a group of $G$ outputs $\{y_1, y_2, \dots, y_G\}$ from the current policy $\pi_{\theta_{\text{old}}}$. The objective maximizes a surrogate loss based on the group-relative advantage. For clarity, we formulate the loss function as follows:

$$\mathcal{J}_{\text{GRPO}}(\theta) = \mathbb{E}_{q \sim \mathcal{D}, \{y_i\} \sim \pi_{\theta_{\text{old}}}} \left[ \frac{1}{G} \sum_{i=1}^{G} \min \left( \rho_i \hat{A}_i, \right. \right.$$
$$\left. \left. \text{clip}(\rho_i, 1 - \epsilon, 1 + \epsilon) \hat{A}_i \right) \right]$$

where $\rho_i = \pi_\theta(y_i|q) / \pi_{\theta_{\text{old}}}(y_i|q)$ denotes the importance sampling ratio, and $\epsilon$ represents the clipping parameter. The advantage $\hat{A}_i$ is computed by normalizing the rewards within the group: $\hat{A}_i = (r_i - \mu)/(\sigma + \delta)$, where $r_i$ represents the total reward for output $y_i$, and $\mu$ and $\sigma$ denote the mean and standard deviation of the reward set $\{r_1, \dots, r_G\}$, respectively.

To operationalize the objective defined in Eq. 2, we design a tailored reward $r(y)$ for GRPO. Specifically, this composite reward integrates three distinct components: format adherence, solution correctness, and structural efficiency.

**(1) Format reward ($r_{\text{fmt}}$).** To guarantee the stability of the hierarchical reasoning structure, we impose strict formatting constraints, such as sequential enumeration. The reward is defined as:

$$r_{\text{fmt}}(y) = \begin{cases} 0, & \text{if } y \text{ adheres to the valid segment format,} \\ -1, & \text{otherwise.} \end{cases}$$

Notably, invalid formats trigger immediate termination of the generation process with a total reward of -1.

**(2) Correctness reward ($r_{\text{acc}}$).** This component verifies the correctness of the derived answer:

$$r_{\text{acc}}(y) = \mathbb{I}(\hat{a} = a^*)$$

where $\mathbb{I}(\cdot)$ denotes the indicator function, evaluating to 1 solely upon an exact match with the ground truth.

*Table 1.* Overall in-distribution and out-of-distribution performance based on Accuracy (Acc.), Token Count (Tok.), and Segmentation Quality (Seg.). We compare CoSMo with various baselines.

| Model | In-Distribution | | | | | | Out-of-Distribution | | | | | | Avg. | | |
|---|---|---|---|---|---|---|---|---|---|---|---|---|---|---|---|
| | HotpotQA | | | Halueval | | | NQ | | | CRAG | | | | | |
| | Acc. | Tok. | Seg. | Acc. | Tok. | Seg. | Acc. | Tok. | Seg. | Acc. | Tok. | Seg. | Acc. | Tok. | Seg. |
| CoT | 81.5 | 138 | 3.6 | 93.0 | 112 | 3.2 | 52.1 | 291 | 7.0 | 48.8 | 327 | 7.6 | 68.9 | 217 | 5.4 |
| *Additive Prompting Strategies* | | | | | | | | | | | | | | | |
| ToT | 81.0 | 436 | 14.6 | 92.4 | 324 | 10.7 | 52.9 | 528 | 15.5 | 49.9 | 615 | 18.0 | 69.1 | 476 | 14.7 |
| HTP | 80.7 | 418 | 12.0 | 87.4 | 365 | 11.4 | 49.4 | 591 | 16.1 | 43.5 | 630 | 16.0 | 65.3 | 501 | 13.9 |
| *Subtractive Prompting Strategies* | | | | | | | | | | | | | | | |
| CoD | 78.0 | **57** | **2.6** | 93.2 | **60** | 2.9 | 53.2 | 149 | 5.3 | 50.2 | 176 | 5.1 | 68.7 | **111** | 4.0 |
| TALE | 81.8 | 153 | 4.6 | 93.5 | 108 | 3.6 | 52.6 | 271 | 7.7 | 49.8 | 268 | 7.3 | 69.4 | 200 | 5.8 |
| *Pruning-based SFT Methods* | | | | | | | | | | | | | | | |
| C3oT | 83.3 | 121 | 2.9 | 90.6 | 108 | 3.0 | 48.3 | 167 | 3.6 | 40.9 | **153** | 3.5 | 65.8 | 137 | 3.3 |
| FS-BoN | 83.5 | 109 | 4.2 | 93.2 | 103 | 4.0 | 50.1 | **147** | 4.7 | 47.2 | 159 | 4.9 | 68.5 | 130 | 4.5 |
| SPIRIT | 80.2 | 104 | 2.8 | 92.0 | 103 | 2.8 | 50.7 | 167 | 4.0 | 45.5 | 186 | 4.2 | 67.1 | 140 | 3.5 |
| *Length-Aligned Reinforcement Learning* | | | | | | | | | | | | | | | |
| LCPO | 84.5 | 110 | 4.1 | 93.0 | 109 | 3.3 | 52.5 | 171 | 4.3 | 50.1 | 189 | 4.3 | 70.0 | 145 | 4.0 |
| ThinkPrune | 83.2 | 97 | 2.9 | 93.1 | 94 | 2.9 | 52.9 | 177 | 3.9 | 49.0 | 182 | 3.9 | 69.6 | 138 | 3.4 |
| *Our Methods* | | | | | | | | | | | | | | | |
| **CoSMo** (Ours) | **88.0** | 135 | **2.6** | **94.0** | 111 | **2.6** | **53.8** | 175 | **3.1** | **50.9** | 189 | **3.1** | **71.7** | 153 | **2.9** |

**(3) Structural efficiency reward ($r_{\text{struct}}$).** This component constitutes the practical realization of our segment-level budgeting strategy. We instantiate the structural cost originally outlined in Eq. 2 as a penalty on segment count deviation beyond a one-step tolerance margin:

$$r_{\text{struct}}(y) = -\max(0, |N - k^*| - 1)$$

The term explicitly penalizes logical depth mismatch beyond this margin, correcting both redundant segments ($N > k^*$) and logical leaps ($N < k^*$). Such invariance to token length grants the policy the flexibility to expand intra-segment reasoning to maximize accuracy ($r_{\text{acc}}$), as long as the structural complexity ($N$) remains aligned with $k^*$.

The final composite reward function $r(y)$ aggregates these components to balance derivation correctness with structural precision:

$$r(y) = r_{\text{fmt}}(y) + r_{\text{acc}}(y) + r_{\text{struct}}(y) \qquad (3)$$

By optimizing this objective, CoSMo effectively guides the model to converge on the optimal reasoning topology without suppressing the deep reasoning capability.

## 5. Experiments

### 5.1. Setups

**Datasets and evaluation metrics.** To rigorously evaluate the effectiveness and robustness of our proposed method, we categorize our experimental benchmarks into In-Distribution (ID) and Out-of-Distribution (OOD) settings. We select HotpotQA (Yang et al., 2018) and Halueval (Li et al., 2023) as the ID datasets to train our model and assess its performance within the source domains. To further probe the model's generalization capability across unseen distributions, we conduct evaluations on Natural Questions (NQ) (Kwiatkowski et al., 2019) and CRAG (Yang et al., 2024) as OOD benchmarks. Performance is measured using three key metrics: accuracy to determine answer correctness, average token length to assess generation efficiency, and average segment count to quantify the granularity of the reasoning chain. A detailed description of these datasets is provided in Appendix A.1.

**Models and baselines.** We evaluate CoSMo against four categories of representative baselines:

(1) **Additive Prompting Strategies**, including CoT (Kojima et al., 2022) for step-by-step reasoning, ToT (Yao et al.,

2023) for tree-structured exploratory search, and HTP (Gui et al., 2025) for divide-and-conquer hypertree planning.

(2) **Subtractive Prompting Strategies**, including CoD (Xu et al., 2025), which instructs the model to restrict reasoning to concise drafts, and TALE (Han et al., 2025), which imposes explicit token-budget constraints.

(3) **Pruning-based SFT Methods**, including C3oT (Kang et al., 2025), which utilizes external LLMs to prune redundant chains; FS-BoN (Munkhbat et al., 2025), which employs Few-Shot Best-of-N selection to favor the shortest valid trajectories; and SPIRIT (Cui et al., 2025), which filters steps based on perplexity thresholds.

(4) **Length-Aligned Reinforcement Learning**, including LCPO (Aggarwal & Welleck, 2025), which applies soft penalties proportional to token consumption, and ThinkPrune (Hou et al., 2025), which implements hard penalties by assigning zero reward for length violations.

We instantiate the above methods using Llama-3.1-8B-Instruct as the backbone model for our main experiments. Results for other backbone models and detailed experimental configurations are provided in Appendix C.1 and Appendix A.3, respectively.

### 5.2. Main Results

As shown in Table 1, CoSMo consistently achieves the lowest average segment count across all datasets. In terms of generation efficiency, it realizes an average token reduction of 29.5% compared to the CoT baseline. Relative to the efficient reasoning methods categorized in blue, CoSMo reduces the average segment count by **28.7%**. Specifically, on the HotpotQA dataset, CoSMo records an average segment count of 2.6, whereas the efficient reasoning baselines average 3.4. Given that over 90% of queries in HotpotQA are 2-hop questions, we analyze the structural redundancy, defined as the number of segments exceeding the ground truth. CoSMo successfully reduces this redundancy from approximately 1.6 segments in CoT and 1.4 segments in efficient baselines to merely 0.6 segments. This corresponds to a reduction in structural bloat of 62.5% and 58.3% respectively, providing compelling evidence of CoSMo's effectiveness in eliminating unnecessary reasoning segments.

In addition to structural compactness, CoSMo consistently maintains superior accuracy across all evaluated datasets. Specifically, compared to all baseline methods, our framework achieves an average improvement of **4.1** points on ID datasets and **2.9** points on OOD datasets. This empirical success validates the core advantage of CoSMo derived from the strategy of refraining from imposing direct constraints on token length to permit deep intra-segment reasoning. By effectively decoupling structural redundancy from reasoning depth, CoSMo significantly enhances model capabilities

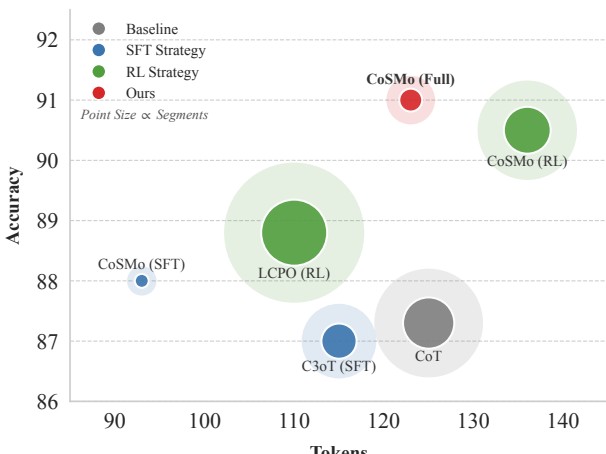

*Figure 3.* Visualization of the efficiency-performance trade-off in the ablation study. The scatter plot illustrates the accuracy of different methods against their token consumption, where the size of each bubble is positively correlated with the number of reasoning segments.

while simultaneously achieving high inference efficiency.

### 5.3. Ablation Study

To rigorously isolate the contributions of individual modules, we conduct a component-wise ablation study. Specifically, we uniformly employ Llama-3.1-8B-Instruct as the base model and conduct validation from two distinct perspectives. For the SFT phase, we compare our $CoSMo_{SFT}$ against the C3oT strategy. For the RL training phase, we contrast our $CoSMo_{RL}$ with the LCPO method. Finally, we present the results of our fully integrated method, $CoSMo_{SFT+RL}$. The remaining experimental setups are consistent with the main experiments, and we report the average metrics across two training datasets. The results are visualized in Figure 3, with detailed numerical data provided in Appendix C.2.

As observed in Figure 3, our $CoSMo_{SFT}$ achieves an average accuracy advantage of 1.0 points over C3oT, while simultaneously reducing token consumption by over 19%. This substantial improvement underscores the effectiveness of our split-merge strategy. Regarding the RL phase, while the LCPO method reduces token consumption, it yields only marginal accuracy gains compared to the baseline and inadvertently increases the number of segments. Conversely, $CoSMo_{RL}$ utilizes fewer segments while significantly enhancing model capability by 3.2 points compared to the CoT baseline. This observation further corroborates the detrimental side effects of redundant reasoning segments on model performance.

Furthermore, the two-stage $CoSMo_{SFT+RL}$ consistently outperforms both $CoSMo_{SFT}$ and $CoSMo_{RL}$ individually, demonstrating that both training stages contribute signifi-

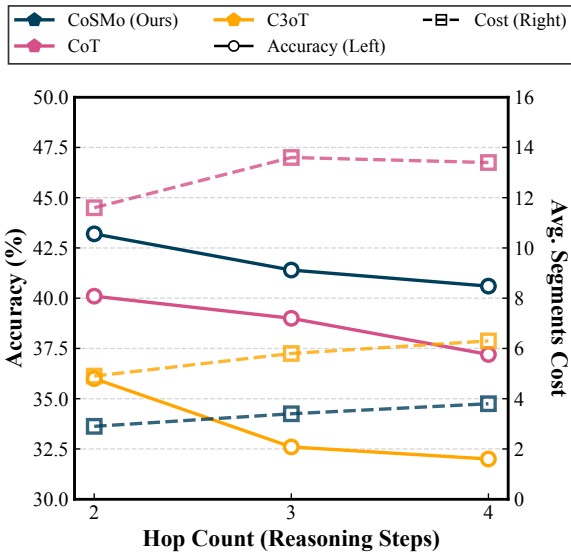

*Figure 4.* Performance comparison of Accuracy (circles, left axis) and Average Segments Cost (squares, right axis) with respect to increasing hop counts.

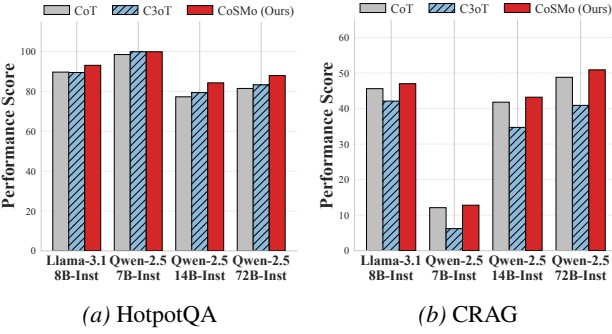

*(a)* HotpotQA    *(b)* CRAG

*Figure 5.* Performance comparison on reasoning benchmarks. We compare CoSMo against CoT and C3oT baselines across different model scales. (a) Results on HotpotQA dataset. (b) Results on CRAG dataset. Our method consistently outperforms baselines.

cantly to the final performance. We find that CoSMo$_{SFT}$ primarily establishes an efficient reasoning paradigm, whereas CoSMo$_{RL}$ focuses on enhancing intrinsic model capabilities. Consequently, the combined framework achieves both high efficiency and response quality, aligning well with recent theoretical insights into post-training dynamics (Zhou et al., 2023; Chu et al., 2025).

### 5.4. Analysis

**Performance across Reasoning Complexities.** To evaluate the robustness of our model against escalating reasoning difficulty, we analyze the performance on the MuSiQue dataset disaggregated by ground-truth reasoning depth (i.e., 2-hop, 3-hop, and 4-hop questions) using the Llama-3.1-8B-Instruct backbone. Note that a higher hop count inherently signifies greater reasoning difficulty. Detailed numerical results are provided in Appendix C.3. As illustrated in Figure 4, we compare CoSMo against CoT and C3oT. In terms of accuracy, CoSMo exhibits remarkable stability as task difficulty escalates, with a marginal decline of only 2.1 points. In sharp contrast, CoT and C3oT suffer more precipitous drops of 3.9 and 4.0 points, respectively. This evidence suggests that the elimination of structural redundancy in CoSMo effectively forestalls the degradation of reasoning capabilities under increased complexity. On the other hand, we observe a precise calibration between the average segments generated by CoSMo and the required reasoning count: as the hop count of the question increases, CoSMo's average segment count rises commensurately, consistently maintaining a deviation of less than 1 from the ground truth.

**Robustness across LLM Judges.** Given the reliance on model-based evaluation, it is critical to ascertain that the performance gains of CoSMo stem from genuine reasoning improvements rather than overfitting to the biases of a specific judge. As illustrated in Figure 5, we evaluate accuracy on representative ID and OOD benchmarks using four distinct high-capacity evaluators, specifically Qwen-2.5-7B-Instruct (Qwen et al., 2025), Qwen-2.5-14B-Instruct, Qwen-2.5-72B-Instruct and Llama-3.1-8B-Instruct. We present detailed experimental data concerning the LLM Judge experiments in Appendix C.4. Irrespective of the chosen LLM judge, CoSMo consistently demonstrates superior accuracy. Specifically, it achieves an average improvement of **+3.9** points on the ID dataset HotpotQA and **+4.5** points on the OOD dataset CRAG. This consistency validates that CoSMo optimizes for intrinsic logical validity and effectively transcends the idiosyncrasies of any single proxy evaluator.

## 6. Conclusion

In this paper, we introduced CoSMo, a framework that redefines reasoning efficiency by explicitly decoupling structural complexity from internal reasoning. Our approach utilizes a synergistic two-stage pipeline comprising a split-merge optimization algorithm that iteratively curates logic-dense data and a structure-aligned reinforcement learning mechanism that aligns reasoning topology with ground-truth complexity. Extensive experiments confirm that CoSMo consistently outperforms state-of-the-art baselines across diverse benchmarks. Particularly on OOD datasets, CoSMo achieves superior elimination of structural redundancy and significant improvements in accuracy. These results strongly demonstrate the robust generalization and broad applicability of CoSMo's efficient reasoning capabilities. Our findings demonstrate the feasibility of eliminating structural redundancy without compromising the semantic depth required for complex deduction, paving the way for future research into the decoupled optimization of efficient LLMs.

## Impact Statement

This work contributes to the advancement of efficient Large Reasoning Models. The primary impact of CoSMo lies in promoting computational sustainability within the field of natural language processing. By strictly aligning the model's reasoning topology with the problem's intrinsic complexity, CoSMo eliminates structural redundancy and avoids the exponential computational cost associated with search-based inference methods (e.g., Tree of Thoughts). Consequently, our approach achieves state-of-the-art performance with a significantly reduced inference budget. This aligns with the broader goal of "Green AI," potentially enabling high-quality reasoning capabilities to be deployed on resource-constrained devices with a lower carbon footprint. Additionally, the resulting segmented reasoning structure inherently improves interpretability, allowing for more transparent analysis of model resource allocation during inference. We do not foresee immediate negative societal consequences specific to this method, though we acknowledge that any advancement in reasoning capabilities carries the general dual-use risk of generating more convincing misinformation.

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

# A. Experimental Details

## A.1. Details of the Datasets

**HotpotQA** (Yang et al., 2018) is a benchmark designed to evaluate multi-step deductive reasoning over separate documents. Unlike traditional datasets that rely on a single context, this dataset necessitates aggregating information from multiple Wikipedia articles to arrive at the correct conclusion. It provides sentence-level ground truth evidence, offering a strong supervision signal that allows for the granular evaluation of a model's reasoning chain. This feature makes it particularly suitable for assessing whether a model can successfully bridge distinct information gaps while maintaining interpretability.

**HaluEval** (Li et al., 2023) is a large-scale collection specifically engineered to detect and mitigate hallucinations in large language models. It comprises a diverse set of generated and human-annotated samples across multiple tasks, including question answering, dialogue, and summarization. The dataset pairs factual responses with hallucinated counterparts, challenging models to discriminate between ground-truth knowledge and plausible but fabricated content. In our setting, it serves as a critical testbed for evaluating the faithfulness of the generated reasoning paths and the model's ability to reject unsupported claims.

**Natural Questions (NQ)** (Kwiatkowski et al., 2019) represents real-world information-seeking behavior, derived from actual anonymized queries submitted to the Google search engine. Each query is paired with a relevant Wikipedia page, requiring the system to read and comprehend the entire document to extract the answer. Unlike synthetic benchmarks, NQ captures the ambiguity and complexity of genuine user intent. It focuses on open-domain question answering, testing the model's capacity to locate precise short answers within long contexts or correctly identify when an answer is absent.

**CRAG (Comprehensive RAG Benchmark)** (Yang et al., 2024) is a dataset designed to assess the robustness of Retrieval-Augmented Generation systems across diverse domains and question types. It addresses the limitations of static knowledge bases by incorporating dynamic and time-sensitive information. The benchmark includes tasks ranging from simple fact retrieval to complex aggregation and comparison. Crucially, it evaluates model performance under varying retrieval qualities—including irrelevant or misleading contexts—making it an essential standard for testing a model's resilience to noise and its effectiveness in selecting useful information segments.

## A.2. Details of the Baselines

We provide detailed descriptions and configuration specifics for the ten baseline methods evaluated in our experiments.

**CoT** (Kojima et al., 2022): Chain-of-Thought facilitates efficient and effective step-by-step reasoning. We implement this by appending the standard trigger phrase "Let's think step by step" to the prompt, guiding the model through a structured linear reasoning process.

**ToT** (Yao et al., 2023): Tree-of-Thought enables multi-path reasoning by guiding large models to generate multiple feasible parallel reasoning paths simultaneously via a trial-and-error search heuristic.

**HTP** (Gui et al., 2025): HyperTree Planning models the reasoning process as a hypertree structure. Building upon standard tree structures, it empowers LLMs with the capability to flexibly apply a divide-and-conquer strategy, decomposing complex queries into a structured hierarchy of sub-tasks.

**CoD** (Xu et al., 2025): Chain-of-Draft explicitly instructs the model to limit its reasoning output to essential points. We implement this using the specific prompt constraint: "Think step by step, but only keep a minimum draft for each thinking step, with 5 words at most."

**TALE** (Han et al., 2025): Token-Aware Level Estimation employs a binary search mechanism to determine an optimal token budget for a given query difficulty. The model is then constrained via the prompt: "Let's think step by step and use less than [Budget] tokens:", where [Budget] is the estimated optimal length.

**C3oT** (Kang et al., 2025): Compressing Chain-of-Thought involves pruning the reasoning traces generated by the backbone model using an external teacher. In our implementation, we uniformly utilize **Qwen-2.5-72B-Instruct** as the external pruner to identify and remove redundant tokens while preserving logical integrity.

**FS-BoN** (Munkhbat et al., 2025): Few-Shot Best-of-N serves as a strong rejection sampling baseline. We adopt a **3-shot** prompting strategy combined with a **Best-of-4** selection policy. Among the sampled responses that lead to the correct answer, we select the shortest one as the refined sample.

**SPIRIT (Cui et al., 2025):** Stepwise Pruning via Iterative Relational Inference filters reasoning steps based on their perplexity. Following the original setting, we define the pruning threshold as $T = \mu - k \cdot \sigma$, where $\mu$ and $\sigma$ denote the mean and standard deviation of the step-level perplexity, respectively. In our experiments, we set $k = 1$.

**LCPO (Aggarwal & Welleck, 2025):** Length-Constrained Policy Optimization applies a soft penalty to the reward function proportional to excess token consumption. In our implementation, we set a soft threshold of 500 tokens. For every additional 100 tokens generated beyond this limit, a penalty of $-0.1$ is deducted from the reward.

**ThinkPrune (Hou et al., 2025):** ThinkPrune imposes a hard upper bound on reasoning length during reinforcement learning. If a response exceeds the limit, it is truncated and assigned a zero reward. We implement a progressive tightening strategy (curriculum learning), where the maximum length constraint is gradually reduced from 2048 tokens down to 512 tokens over the course of training.

### A.3. Details of Training

We provide comprehensive implementation details regarding our training pipeline. All experimental runs were executed on NVIDIA A100 GPUs. Specifically, the computational infrastructure comprised 4 A100 GPUs for the Reinforcement Learning (RL) phase and 2 A100 GPUs for the Supervised Fine-Tuning (SFT) phase. For SFT, we employed Low-Rank Adaptation (LoRA) to facilitate parameter-efficient fine-tuning. The global batch size was maintained at 64, utilizing a micro-batch size of 2 per device. We adopted a learning rate of $2 \times 10^{-5}$ over a duration of 3 epochs, retaining the final checkpoint for subsequent optimization stages.

Regarding the split-merge optimization algorithm, we consistently utilized the Qwen-2.5-72B-Instruct model for both the consistency judge $\mathcal{M}_{\text{judge}}$ and the semantic generator $\mathcal{M}_{\text{gen}}$. Additionally, we set the maximum iteration limit $T_{\text{max}}$ to 5. Specifically, the prompts used for the split and merge operations during data curation are detailed in Appendix E.

During the RL phase, both training and validation batch sizes were set to 64. We configured the GRPO algorithm with a group size of 8 and a clipping parameter of 0.1. For the reward function, we employed an unweighted summation strategy. Furthermore, we incorporated a tolerance margin of 1 step into the structural efficiency reward. This adjustment allows the generated segment count $N$ to deviate from the ground truth $k^*$ by $\pm 1$, thereby accommodating acceptable variations in reasoning granularity.

### A.4. Details of Reasoning

In this section, we delineate the specific protocols governing the inference process. We instruct the model to generate outputs adhering to a strict structured format. Reasoning segments are extracted via rule-based parsing of the generated numbered lists. Answer correctness is subsequently validated using an LLM-based Judge. The detailed prompt templates governing this generation and evaluation process are provided in Appendix E.

## B. More Related Work

Beyond the text-based logical reasoning tasks discussed in the main text, such as improving the complex reasoning of large language models via instantiated multi-step synthetic logical data (Wang et al., 2026), the core methodologies of CoSMo share connections with recent advancements in other AI domains. Specifically, these methodologies include identifying structural redundancies for efficient inference and employing reinforcement learning to navigate complex topological spaces.

**Efficient Inference in Vision-Language-Action Models.** Identifying and eliminating redundant computational steps is increasingly vital in multimodal and embodied AI systems, where maintaining high efficiency without sacrificing semantic alignment is a major challenge. For instance, recent works like Evo-1 (Lin et al., 2025) explore lightweight architectures to preserve semantic alignment during the execution of vision-language-action (VLA) models. Similarly, to tackle computational bottlenecks from dense visual inputs, VLA-Pruner (Liu et al., 2025b) introduces a temporal-aware dual-level visual token pruning technique. These approaches align with CoSMo's motivation to intelligently discard less informative tokens or states to accelerate inference, achieving a better trade-off between computational efficiency and complex task-solving capabilities.

**Reinforcement Learning for Complex Structural Optimization.** In our framework, we employ reinforcement learning (RL) to navigate the structural space of reasoning topologies. This approach builds on the broader success of RL in solving

*Table 2.* Overall in-distribution and out-of-distribution performance based on Accuracy (Acc.), Token Count (Tok.), and Segmentation Quality (Seg.) with Qwen-2.5-7B-Instruct.

| Model | In-Distribution | | | | | | Out-of-Distribution | | | | | | Avg. | | |
|---|---|---|---|---|---|---|---|---|---|---|---|---|---|---|---|
| | HotpotQA | | | Halueval | | | NQ | | | CRAG | | | | | |
| | Acc. | Tok. | Seg. | Acc. | Tok. | Seg. | Acc. | Tok. | Seg. | Acc. | Tok. | Seg. | Acc. | Tok. | Seg. |
| CoT | 83.7 | 127 | 3.0 | 92.4 | 117 | 3.3 | 54.5 | 158 | 3.3 | 51.3 | 178 | 3.6 | 70.5 | 145 | 3.3 |
| *Additive Prompting Strategies* | | | | | | | | | | | | | | | |
| ToT | 77.6 | 240 | 7.8 | 90.6 | 244 | 8.5 | 53.7 | 253 | 7.9 | 51.9 | 258 | 7.7 | 68.5 | 249 | 8.0 |
| HTP | 81.1 | 479 | 11.4 | 91.1 | 408 | 11.2 | 53.4 | 545 | 12.3 | 45.7 | 628 | 13.6 | 67.8 | 515 | 12.1 |
| *Subtractive Prompting Strategies* | | | | | | | | | | | | | | | |
| CoD | 81.0 | **66** | 3.1 | 91.5 | **56** | 3.3 | 53.6 | **74** | 3.4 | 51.8 | **93.1** | 3.7 | 69.5 | **72** | 3.4 |
| TALE | 79.5 | 102 | 3.1 | 90.8 | 89 | 3.1 | 56.2 | 160 | 3.5 | 50.5 | 178 | 3.8 | 69.3 | 132 | 3.4 |
| *Pruning-based SFT Methods* | | | | | | | | | | | | | | | |
| C3oT | 81.7 | 134 | **1.8** | 92.6 | 108 | **2.8** | 47.9 | 136 | **2.2** | 43.1 | 166 | 3.7 | 66.3 | 136 | **2.6** |
| FS-BoN | 83.9 | 99 | 4.0 | 92.8 | 90 | 4.2 | 51.3 | 129 | 4.2 | 48.5 | 146 | 4.0 | 69.1 | 116 | 4.1 |
| SPIRIT | 79.1 | 98 | 2.8 | 90.6 | 76 | 2.9 | 51.1 | 144 | 3.7 | 48.3 | 169 | 3.8 | 67.3 | 122 | 3.3 |
| *Length-Aligned Reinforcement Learning* | | | | | | | | | | | | | | | |
| LCPO | 83.8 | 101 | 3.9 | 93.0 | 97 | 4.0 | 54.9 | 156 | 4.2 | **52.8** | 170 | 4.1 | 71.1 | 131 | 4.1 |
| ThinkPrune | 83.1 | 99 | 2.8 | 92.6 | 92 | 3.1 | 53.7 | 153 | 3.8 | 52.1 | 166 | 3.8 | 70.4 | 128 | 3.4 |
| *Our Methods* | | | | | | | | | | | | | | | |
| **CoSMo** (Ours) | **89.1** | 108 | 2.8 | **94.9** | 99 | **2.8** | **56.2** | 148 | 3.0 | 52.5 | 158 | **2.9** | **73.9** | 128 | 2.9 |

multi-stage combinatorial optimization problems, particularly in Electronic Design Automation (EDA). For example, RL is now widely used in AI-based chip placement algorithms, leading to comprehensive benchmarking efforts that evaluate end-to-end performance (Wang et al., 2024). Advanced methods integrate RL with Monte Carlo tree search to accelerate macro placement (Geng et al., b). Furthermore, frameworks like LaMPlace (Geng et al., a) demonstrate that RL agents can effectively optimize cross-stage metrics in macro placement. These applications highlight the capability of RL to explore and optimize complex structural configurations, which shares the same objective as CoSMo: discovering the optimal topological graph for mathematical and logical reasoning.

# C. More Results

## C.1. Detailed Results of Main Results

In this section, we present the results using Qwen-2.5-7B-Instruct as the backbone model in Table 2 to substantiate the universality of our approach across different model architectures. These results corroborate the findings presented in the main text and demonstrate that the efficacy of CoSMo is not limited to specific model architectures.

## C.2. Detailed Results of Ablations

In this section, we provide the complete table regarding the ablation study, as shown in Table 3. It highlights the specific contributions of the split-merge algorithm and the segment-level budget to the overall performance gains.

## C.3. Detailed Results on Reasoning Complexity

We present the specific experimental data regarding the robustness analysis in Table 4. These metrics offer a deeper insight into how the reasoning structure adapts to varying hop counts.

*Table 3.* Ablation study of the **CoSMo** framework. We analyze SFT and RL as **orthogonal** directions. The middle sections evaluate SFT and RL policies independently applied to the base model. The bottom section shows the performance of our full framework combining the best SFT and RL strategies.

| Method | HotpotQA | | | HaluEval | | | Average | | |
|---|---|---|---|---|---|---|---|---|---|
| | Acc. | Tok. | Seg. | Acc. | Tok. | Seg. | Acc. | Tok. | Seg. |
| CoT Baseline | 81.5 | 138 | 3.6 | 93.0 | 112 | 3.2 | 87.3 | 125 | 3.4 |
| *Analysis I: SFT Strategies* | | | | | | | | | |
| + C3oT | 83.3 | 121 | 2.9 | 90.6 | 108 | 3.0 | 87.0 | 115 | 3.0 |
| + CoSMo$_{SFT}$ | 84.7 | **98** | **2.1** | 91.2 | **87** | **2.3** | 88.0 | **93** | **2.2** |
| *Analysis II: RL Alignment* | | | | | | | | | |
| + LCPO | 84.5 | 110 | 4.1 | 93.0 | 109 | 3.3 | 88.8 | 110 | 3.7 |
| + CoSMo$_{RL}$ | 87.4 | 145 | 3.2 | 93.5 | 126 | 3.3 | 90.5 | 136 | 3.3 |
| **CoSMo$_{SFT+RL}$ (Ours)** | **88.0** | 135 | 2.6 | **94.0** | 111 | 2.6 | **91.0** | 123 | 2.6 |

*Table 4.* Detailed performance and average segment counts across different reasoning depths on MuSiQue.

| Method | Accuracy (%) | | | Avg. Segments | | |
|---|---|---|---|---|---|---|
| | 2-hop | 3-hop | 4-hop | 2-hop | 3-hop | 4-hop |
| CoT | 40.1 | 39.0 | 36.2 | 11.6 | 13.6 | 13.4 |
| C3oT | 36.0 | 32.6 | 32.0 | 4.9 | 5.8 | 6.3 |
| **CoSMo (Ours)** | **43.2** | **41.4** | **41.1** | **2.9** | **3.4** | **3.8** |

*Table 5.* Robustness evaluation of reasoning accuracy across different LLM Judges. To mitigate the potential bias of a single evaluator, we report the Accuracy (Acc) assessed by Llama-3.1-8B-Inst, Qwen-2.5-7B-Inst, Qwen-2.5-14B-Inst, and Qwen-2.5-72B-Inst. **CoSMo** demonstrates consistent superiority regardless of the judge employed. For each position in the table, we place the HotpotQA data first and the CRAG data second.

| Method | Llama-3.1-8B-Inst | Qwen-2.5-7B-Inst | Qwen-2.5-14B-Inst | Qwen-2.5-72B-Inst |
|---|---|---|---|---|
| | Acc (↑) | Acc (↑) | Acc (↑) | Acc (↑) |
| CoT (Kojima et al., 2022) | 89.7/45.6 | 98.5/12.1 | 77.3/41.8 | 81.5/48.8 |
| C3oT (Kang et al., 2025) | 89.5/42.1 | 99.9/6.2 | 79.4/34.7 | 83.3/40.9 |
| **CoSMo (Ours)** | 93.1/47.0 | 99.9/12.8 | 84.3/43.2 | 88.0/50.9 |

## C.4. Detailed Results on LLM Judges

In this section, we present detailed performance metrics across different LLM evaluators to substantiate the robustness of CoSMo. Table 5 enumerates the accuracy scores for CoSMo and baseline methods on HotpotQA and CRAG when evaluated by Qwen-2.5-7B-Instruct, Qwen-2.5-14B-Instruct, Qwen-2.5-72B-Instruct, and Llama-3.1-8B-Instruct. The results demonstrate that our method maintains its lead across varying evaluation protocols. Furthermore, we observe that CoSMo consistently outperforms baselines regardless of the strictness or bias of the specific judge employed.

## D. Case Study

In this section, we present a concrete example from the dataset to illustrate how CoSMo transforms a reasoning chain through the Split-Merge optimization process.

**Problem Instance**

**Question:** The 1997 American romantic comedy with Matt Dillon, Debbie Reynolds, etc. stars which American actress who also voiced which role in the "Toy Story" franchise?

**Stage 1: Initial Generation (Raw Output)**

*Observation: The initial chain contains redundant verification steps (e.g., checking Debbie Reynolds' bio unnecessarily in Step 3, verifying Matt Dillon's voice roles in Step 4) and inefficient search attempts.*

> I will first analyze the question to identify the specific criteria for the film: It is a 1997 American romantic comedy, and it stars Matt Dillon and Debbie Reynolds. I need to verify the film's title. I will scan the document titles. Document 'In & Out (film)' mentions "In & Out is a 1997 American romantic comedy film... starring... Matt Dillon... and Debbie Reynolds". Now I will perform a redundant check on Debbie Reynolds to ensure she fits the description. I see in Document 'Susan Slept Here' she is an actress, and Document 'The Unsinkable Molly Brown (film)' confirms she is a famous American actress. This confirms the film identification is likely correct. Now I need to check the other actors in 'In & Out' to see if they are in "Toy Story". I will first check Matt Dillon. I am checking if Matt Dillon voiced anyone in "Toy Story". There is no mention of Matt Dillon in the "Toy Story" documents. Next, I will check the cast list of 'In & Out' again. It lists Joan Cusack. I will now investigate Joan Cusack. I will check Document 'Joan Cusack'. I need to verify two things about Joan Cusack. First, was she in 'In & Out'? Yes, the document says she received an Academy Award nomination for "In & Out". Second, did she voice a role in the "Toy Story" franchise? The same document says she is "known as the voice of Jessie in the 'Toy Story' franchise". I will double-check the character name. Document 'Toy Story 2' confirms "Jessie (voiced by Joan Cusack)". All checks pass. \boxed{Joan Cusack voiced the character Jessie in the "Toy Story" franchise.}

**Stage 2: After Merge Phase (Over-Compression)**

*Observation: The logic is compressed into a "shortcut." While the answer is correct, the reasoning leaps from identifying the film directly to the answer without explicitly citing the bridging evidence for the actress, violating the explicit reasoning requirement (Logical Leap).*

> Based on the prompt asking for a 1997 romantic comedy with Matt Dillon and Debbie Reynolds, I can quickly identify the film is 'In & Out' from the provided documents. Since I know the film is 'In & Out', I can immediately see from the 'Joan Cusack' document that she stars in that film and is famously the voice of Jessie in "Toy Story". The connection is obvious and requires no further search steps.
>
> \boxed{Joan Cusack voiced the character Jessie in the "Toy Story" franchise. }

**Stage 3: Final Output (After Split Refinement)**

*Observation: The split operator detects the logical gap in Stage 2 and expands it into a coherent, verifiable sequence. This structure aligns perfectly with the problem's multi-hop nature (Film ID → Actress ID → Role Verification → Synthesis).*

> **Identify the film:** According to Document 'In & Out (film)', the movie described as a "1997 American romantic comedy film" starring "Matt Dillon" and "Debbie Reynolds" is "In & Out". **Identify the target actress:** I need to find an actress from "In & Out" who is in the "Toy Story" franchise. According to Document 'Joan Cusack', she appeared in "In & Out" (earning an Academy Award nomination) and is explicitly identified as "the voice of Jessie in the 'Toy Story' franchise". **Verify the role context:** Document 'Toy Story 2' provides supporting evidence, listing "Jessie (voiced by Joan Cusack)" as a character introduced in the sequel. **Synthesize the answer:** The actress is Joan Cusack, and the role is Jessie.
>
> \boxed{Joan Cusack voiced the character Jessie in the "Toy Story" franchise. }

# E. Prompt Templates

To ensure reproducibility, we provide the exact prompt templates used for the structured generation phase and the answer judgment phase in Figure 7 and Figure 6, respectively. In addition, we present the prompt templates for the merge and split operations in Figure 8 and Figure 9, respectively.

**Role & Objective**

You are an expert evaluator tasked with judging model predictions against a ground truth. You must determine if a prediction is correct based on a strict set of verification rules.

**Verification Rules**

1. **Gold Standard Assumption:** We postulate that the provided ground truth serves as the absolute reference for correctness.

2. **Criteria for Success (Score = 1):** A prediction is deemed correct if it satisfies any of the following conditions:

   - *Strict Identity:* The output is legally identical to the reference answer.
   - *Value Precision:* For numerical answers, the output falls within a negligible error margin of the reference value.
   - *Semantic Condensation:* The output successfully captures the core information of the reference in a summarized form.
   - *Set Equality:* When the answer is a collection, the output contains the exact same elements as the reference, disregarding order.

3. **Failure Modes (Score = 0):** The prediction receives a zero score if it exhibits:

   - *Logical Inconsistency:* The reasoning contains internal conflicts or self-contradictions.
   - *Non-Responsiveness:* The content deviates entirely from the query's intent.
   - *Unmatched Default:* Any output failing to meet the success criteria falls into this category.

---

**Input Data:**

- **Question:** {question}

- **Ground Truth:** {ground_truth_answer}

- **Prediction:** [Model Output]

*Figure 6.* The structured inference prompt utilized in LLM judge.

**Role & Objective**

You are a sophisticated reasoning engine designed to solve complex queries by synthesizing information from provided references. Your goal is to produce a logically sound derivation that leads to a precise final answer.

**Operational Protocols**

1. **Atomic Reasoning Units:** Every segment in your reasoning process must represent a complete logical inference. You must consolidate document retrieval, evidence citation, and factual deduction into single, substantial segments. Do not fragment a single thought into multiple segments.

2. **Sequential Structure:** The reasoning process must be articulated as a strictly numbered list (1., 2., 3., ...), ensuring a clear linear flow of logic.

3. **Source Fidelity:** All answers must be grounded in the factual information contained within the provided References or established knowledge.

---

**Format Requirements**

1. **Reasoning Chain:**
   Initiate your response with the numbered reasoning segments derived from the protocols above.

2. **Final Answer Encapsulation:**
   The final answer must be strictly encapsulated within \boxed{}. No other text should surround the box in the final line.

**Required Output Template:**

1. [First complete logical inference segment]
2. [Second complete logical inference segment]
. . .
\boxed{Your Final Answer}

---

**Input Data:**

- **Question:** {question}

- **References:** {references}

*Figure 7.* The structured inference prompt utilized in CoSMo.

**Role & Objective**
You are a logical editor operating in a strict JSON-output mode. Your task is to analyze two reasoning steps for semantic and logical redundancy.

**Task Description**
Evaluate whether `Reasoning Step 1` and `Reasoning Step 2` belong to the same atomic logical inference unit based on the provided guidelines.

**Operational Logic**

1. **Merge Condition:** If the steps represent a single, continuous thought process or if the second step is merely a paraphrase/trivial extension of the first, merge them. Refine the merged text to be one concise sentence.

2. **Split Condition:** If the steps represent distinct logical actions (e.g., retrieving a document vs. deducing a new fact), keep them separate. Refine each step individually to be clear and concise.

---

**Output Format (JSON Only)**

```
{
  "decision": "merge" | "split",
  "merged_text": "string (required if decision is merge)",
  "refined_1": "string (required if decision is split)",
  "refined_2": "string (required if decision is split)"
}
```

---

**Input Data:**

- **Reasoning Step 1:** {s1}

- **Reasoning Step 2:** {s2}

- **Rules:** {instruction}

*Figure 8.* The prompt template used for the Merge Phase in the Split-Merge Optimization algorithm.

**Role & Objective**
You are a logical editor operating in a strict JSON-output mode. Your task is to decompose a complex reasoning step into two distinct atomic steps.

**Task Description**
Analyze the given `Input Reasoning` step. If it contains multiple distinct logical actions (e.g., information retrieval followed by a deduction), break it down into two separate, concise steps.

**Operational Protocols**

1. **Atomic Decomposition:** The input step must be split into exactly two granular steps.

2. **Logical Flow:** Ensure that Step 1 logically precedes Step 2.

3. **Factuality:** Do not add new facts. The split must rely solely on the information present in the input.

---

**Output Format (JSON Only)**

```
{
  "step_1": "string",
  "step_2": "string"
}
```

---

**Input Data:**

- **Input Reasoning:** {reasoning_text}

- **Rules:** {instruction}

*Figure 9.* The prompt template used for the Split Phase in the Split-Merge Optimization algorithm.

