# OpenReview forum: "Short Chains, Deep Thoughts: Balancing Reasoning Efficiency and Intra-Segment Capability via Split-Merge Optimization"
_ICML.cc/2026/Conference — ICML 2026 regular_

### Official Review · Reviewer_AdJP · 2026-03-01

**Soundness:** 2
**Presentation:** 3
**Significance:** 2
**Originality:** 3
**Overall Recommendation:** 4
**Confidence:** 3

**Summary:**

This paper addresses reasoning efficiency in Large Reasoning Models by optimizing the structure of reasoning chains instead of reducing token length. The authors propose CoSMo, a framework that models reasoning as a sequence of segments and refines reasoning chains through split and merge operations to remove redundancy and improve coherence. A structure-aligned reinforcement learning objective with a segment-level budget encourages efficient reasoning during training. Experiments on multi-hop QA benchmarks show that CoSMo improves accuracy while reducing the number of reasoning segments compared to existing methods.

**Compliance With Llm Reviewing Policy:**

Affirmed.

**Final Justification:**

Thanks for your rebuttal! I upped my score.

**Key Questions For Authors:**

see the weaknesses

**Limitations:**

yes, in the `impact statement`

**Strengths And Weaknesses:**

### Strengths
1. Clear motivation and findings. Improving reasoning efficiency is an important topic; the work focuses on segment-level redundancy beyond token-level. And the findings in Sec. `3.2` are insightful.
2. Good writing and presentation. Writing is clear, easy to get the point.


### Weaknesses
1. I agree that the finding (Sec. `3.2`) is insightful, but would there be any violation, for example, are there cases where split-merge operations harm reasoning performance (merging segments that contain subtle logical distinctions)?
2. For evaluations, there should include more popular reasoning benchmarks (GSM8K, MATH500, ARC-C...) and compare with more efficient reasoning methods via RL (*e.g.*, Thinkless [NeurIPS'25]).
3. The authors can consider adding a discussion about CoSMo's potential in visual reasoning (Does the segment still work for multimodal reasoning?) References: RewardMap [ICLR'26], Seg-Zero, PixelThink...

---

> ### Author Rebuttal · Authors · 2026-03-30
>
> We thank the reviewer for the insightful, valuable, and positive comments. We address the concerns in detail as follows. We sincerely hope that our response could properly address your concerns.
> ### Weakness 1
> >I agree that the finding (Sec. 3.2) is insightful, but would there be any violation, for example, are there cases where split-merge operations harm reasoning performance (merging segments that contain subtle logical distinctions)?
>
> We highly appreciate this insightful question. Indeed, mistakenly merging segments that contain subtle logical distinctions is the most frequent error phenomenon within the CoSMo algorithm.
>
> To mitigate the potential harm to reasoning performance caused by such violations, our framework relies on two key mechanisms. First, **because our optimization algorithm is iterative, early misjudgments can often self-heal in subsequent iterations**. Furthermore, a crucial design choice of our algorithm is that it strictly relies on merging rather than pruning. Consequently, **even if a critical reasoning step is mistakenly identified as redundant, its semantic content is integrated and preserved in the final output rather than being permanently discarded**. This ensures that the essential logical components are safely retained, minimizing any negative impacts on final accuracy.
>
> We will explicitly incorporate a detailed discussion regarding these algorithm failure modes and the robustness of the split-merge operations into the revised manuscript.
> ### Weakness 2
> >For evaluations, there should include more popular reasoning benchmarks (GSM8K, MATH500, ARC-C...) and compare with more efficient reasoning methods via RL (e.g., Thinkless).
>
> We sincerely agree that evaluating on popular mathematical reasoning benchmarks and comparing with recent advanced RL-based efficient methods significantly strengthens our paper.
>
> To address this, **we conducted additional experiments on the GSM8K and MATH500 datasets**. Consistent with our main experiments, we utilized Llama-3.1-8B-Instruct as the backbone model. We trained the models on the 7.5k GSM8K training dataset. For the baseline Thinkless, we strictly reproduced its DeGRPO training process. For CoSMo, to ensure a rigorously fair comparison, we exclusively applied our Structure-Aligned RL phase. We standardized the training epoch to 1, with all other experimental configurations remaining identical to our main setup.
>
> The evaluation results on the GSM8K test set and the OOD MATH500 dataset are summarized below:
> Method | GSM8k (Acc/Tok/Seg) | MATH500 (Acc/Tok/Seg)
> -|:-:|:-:
> CoT    | 80.7/189/4.3 | 45.0/561/11.7
> Thinkless   | 83.5/121/4.0 | 46.8/435/8.9
> CoSMo  | 85.9/149/3.8 | 47.0/276/4.6
>
> As demonstrated in the table, **CoSMo consistently outperforms Thinkless in terms of accuracy on both datasets**. Furthermore, when evaluating on the more complex OOD MATH500 dataset, the structural compression achieved by Thinkless is relatively limited. **In contrast, CoSMo exhibits significantly stronger generalization capabilities, successfully compressing the reasoning topology to 4.6 segments and 276 tokens**.
>
> We will explicitly include these experimental results in the revised manuscript.
> ### Weakness 3
> >The authors can consider adding a discussion about CoSMo's potential in visual reasoning (Does the segment still work for multimodal reasoning?) References: RewardMap, Seg-Zero, PixelThink...
>
> We sincerely thank the reviewer for this highly insightful suggestion and have carefully studied the suggested references (Seg-Zero, RewardMap, and PixelThink). We completely agree that extending CoSMo to visual reasoning is a promising direction, and our segment-level optimization philosophy remains highly applicable.
>
> **CoSMo's core objective of eliminating reasoning redundancy aligns perfectly with the reasoning backbones of Multimodal Large Language Models (MLLMs)**. The deployment philosophy remains consistent by elevating the split and merge operations to the multimodal dimension. Specifically, the model can merge redundant logical segments (e.g., redundant descriptions of identical visual features) and split logically broken segments (e.g., skipping necessary intermediate visual analysis steps).
>
> Regarding concrete training implementations, **datasets like REASONMAP-PLUS and ReasonSeg-DIFF from these works are exceptionally ideal for training CoSMo**. For instance, a task like "finding a fish with an open mouth" requires explicitly structured logical steps: locating the fish's head, identifying the mouth contour, and distinguishing open versus closed features. Such fine-grained, step-by-step supervision is perfectly suited to guide the structural optimization of reasoning chains during our split-merge and RL phases.
>
> In the revised manuscript, we will add a dedicated discussion on CoSMo's potential in multimodal reasoning and formally incorporate these pioneering works as strong baselines.

---

> > ### Author Rebuttal · Reviewer_AdJP · 2026-04-03
> >
> > Thanks for your rebuttal. I have some further questions for W1. Can you provide some details (e.g., examples and evaluation metrics...) to further illustrate this frequent error phenomenon?

---

> > > ### Author Response · Authors · 2026-04-04
> > >
> > > We sincerely thank the reviewer for the prompt feedback and the opportunity to further clarify this aspect of our algorithm. To comprehensively address your question, we provide the corresponding evaluation metrics and concrete examples below.
> > >
> > > 1. Quantitative Evaluation Metrics:
> > >
> > > To systematically quantify this error phenomenon, we conducted a rigorous manual evaluation. **We randomly sampled 100 split operations and 100 merge operations generated during the data curation phase on the HotpotQA dataset**. Human experts then verified the logical correctness of each operation.
> > >
> > > **The evaluation results show that the accuracy of merge operations is 89%, and the accuracy of split operations reaches 95%**. This demonstrates that while erroneous operations (such as over-compressing subtle logical distinctions) do occur, their frequency is quite low. Therefore, the overall error phenomenon is highly controllable and well-managed within the CoSMo framework.
> > >
> > > 2. Concrete Examples of Errors:
> > >
> > > To concretely illustrate what these errors look like when they do occur, we provide examples of both an erroneous merge and an erroneous split.
> > >
> > > Example A: Erroneous Merge (Logical Over-compression)
> > >
> > > Original Segments:
> > >
> > > $s_i$: "I need to verify the film's title. I will scan the document titles. Document 'In & Out (film)' mentions that In & Out is a 1997 American romantic comedy film directed by Frank Oz and starring Kevin Kline, Tom Selleck, Joan Cusack, Matt Dillon, and Debbie Reynolds."
> > >
> > > $s_{i+1}$: "I will now investigate Joan Cusack. Document 'Joan Cusack' says she received an Academy Award nomination for 'In & Out' and is known as the voice of Jessie in the Toy Story franchise."
> > >
> > > Erroneously Merged Segment ($s_{\text{new}}$): "Based on the prompt, the 1997 film is 'In & Out', and Joan Cusack, who stars in it, voices Jessie in Toy Story."
> > >
> > > Analysis: The merge completely deletes the intermediate verification step. It fuses the identification of the film directly with the identification of the actress and her voice role, destroying the sequential logical premise.
> > >
> > > Example B: Erroneous Split (Logical Fragmentation)
> > >
> > > Original Segment:
> > >
> > > $s_i$: "From Document 'Colin Blakely', we learn he was a Northern Irish character actor who was nominated for a BAFTA Award for the Academy Award-nominated film 'Equus'."
> > >
> > > Erroneously Split Segments:
> > >
> > > $s_{i\_a}$: "Let me look at another actor associated with Equus. From Document 'Colin Blakely', we learn he was a Northern Irish character actor."
> > >
> > > $s_{i\_b}$: "The document explicitly mentions that he was nominated for a BAFTA Award for his work in the Academy Award-nominated film 'Equus'."
> > >
> > > Analysis: A single, cohesive logical step (identifying the actor and their role in the target film) is artificially shattered into micro-steps, introducing a disjointed flow that breaks the natural chain of thought without adding structural clarity.
> > >
> > > We will explicitly incorporate the above discussion, evaluation metrics, and concrete examples into the revised manuscript.

---

### Official Review · Reviewer_5AEd · 2026-03-05

**Soundness:** 3
**Presentation:** 2
**Significance:** 2
**Originality:** 2
**Overall Recommendation:** 4
**Confidence:** 4

**Summary:**

CoSMo optimizes reasoning efficiency in large reasoning models by decoupling segment count from intra-segment reasoning depth. It uses split-merge refinement toward ground-truth hop counts, followed by segment-level RL.

**Compliance With Llm Reviewing Policy:**

Affirmed.

**Final Justification:**

I have read the author's response and raised the final score. Good luck!

**Key Questions For Authors:**

1. Judge reliability: any inter-judge or human validation?
2. Can segment-level effect be isolated from RL algorithm choice?
3. Any evidence CoSMo works outside QA tasks?

**Limitations:**

Yes.

**Strengths And Weaknesses:**

# Strengths

- Clear idea: decoupling segment structure from token-level reasoning.
- Split-merge algorithm is simple and effective.
- Two-stage pipeline (SFT + RL) is reasonable and supported by results.

# Weaknesses

- Heavy reliance on external LLM for data curation; errors may propagate.
- Assumes ground-truth hop counts are optimal; may suppress valid longer chains.
- Baseline comparisons confounded by different RL algorithms.
- Evaluation uses LLM judges with high variance; reliability unclear.
- Limited analysis of failure modes or generalization beyond QA.

---

> ### Author Rebuttal · Authors · 2026-03-30
>
> We thank the reviewer for the insightful, valuable, and positive comments. We address the concerns in detail as follows. We sincerely hope that our response could properly address your concerns. If so, we would deeply appreciate it if you could raise your score. If not, please let us know your further concerns, and we will continue actively responding to your comments and improving our submission.
>
> ### Weakness 1
> >Heavy reliance on external LLM for data curation; errors may propagate.
>
> We appreciate the concern; please refer to **Weakness 2 of Reviewer bBT7**.
> ### Weakness 2
> >Assumes ground-truth hop counts are optimal; may suppress valid longer chains.
>
> We highly appreciate this concern. In fact, preventing the suppression of valid reasoning chains is the exact motivation behind the design of CoSMo.
>
> First, **assuming optimal ground-truth hop counts is highly reasonable for multi-hop QA, where the logical path is typically definitive**. This premise is also strongly corroborated by our preliminary findings in Section 3.2. Consequently, if a model's generated hop count strictly exceeds the ground truth k*, it almost certainly indicates the presence of completely redundant or hallucinatory reasoning steps, which should be penalized.
>
> More importantly, **to ensure that valid deep reasoning is not suppressed, CoSMo strictly refrains from imposing any constraints on intra-segment length**. By allowing comprehensive logical articulation within each necessary step, CoSMo explicitly permits and encourages the model to generate deep, valid reasoning.
> ### Weakness 3 & Question 2
> >Baseline comparisons confounded by different RL algorithms.
>
> >Can segment-level effect be isolated from RL algorithm choice?
>
> We respectfully clarify this point. Specifically, **all RL-based baselines (including LCPO and ThinkPrune) and our CoSMo framework were implemented using the exact same GRPO algorithm**. The only distinction between these experimental runs was the specific reward design dictated by each method. Therefore, any performance improvements observed can be solely attributed to the segment-level effect. We will ensure this experimental control is explicitly stated in the final manuscript.
> ### Weakness 4 & Question 1
> >Evaluation uses LLM judges with high variance; reliability unclear.
>
> >Judge reliability: any inter-judge or human validation?
>
> We address this concern through three aspects:
>
> First, we clarify our rationale for adopting LLM judges. In complex QA tasks, traditional metrics like Exact Match (EM) suffer from severe biases, frequently misjudging semantically correct answers due to minor formatting variations. Consequently, employing LLM-based evaluation has become an established trend to ensure fair assessment.
>
> Second, we selected the Qwen-2.5-72B-Instruct model as our evaluator. This choice is strongly supported by the recent comprehensive study "LLM-as-a-Judge: Reassessing the Performance of LLMs in Extractive QA". Evaluating massive QA samples, the study concludes that Qwen-2.5-72B achieves the highest alignment with human judgment, outperforming alternatives like Llama-3.3-70B. Most importantly, **when evaluating outputs from Llama-3.1-8B-Instruct (the primary backbone in our main experiments), Qwen-2.5-72B demonstrated an exceptional Pearson correlation coefficient of 0.945**.
>
> Finally, to empirically guarantee the reliability of our evaluation pipeline, we conducted a rigorous manual human evaluation. We sampled 100 reasoning traces from our four evaluated datasets and compared the LLM judge's verdicts against human annotations. **We found a remarkable 95% consistency rate between the LLM judge and human evaluators**. This provides concrete evidence that our evaluation protocol is highly reliable and exhibits minimal variance.
> ### Weakness 5 & Question 3
> >Limited analysis of failure modes or generalization beyond QA.
>
> >Any evidence CoSMo works outside QA tasks?
>
> We agree that analyzing failure modes and broader generalization is highly valuable and will incorporate both into the revision.
>
> Regarding failure modes, the primary source of error in the CoSMo algorithm arises from the reasoning granularity judge (i.e., misjudging whether a segment should be split or merged). However, **because our optimization algorithm is iterative, early misjudgments can often self-heal in subsequent iterations**. Furthermore, a crucial design choice of our algorithm is that it strictly relies on merging rather than pruning. Consequently, **even if a critical reasoning step is mistakenly identified as redundant, its semantic content is integrated and preserved in the final output rather than being permanently discarded**. This significantly mitigates the risk of catastrophic logical breaks and error propagation.
>
> Regarding generalization beyond QA, we have conducted additional experiments on mathematical reasoning datasets (GSM8k and MATH500), please refer to our detailed response to **Weakness 2 of Reviewer AdJP**.

---

> > ### Author Rebuttal · Reviewer_5AEd · 2026-04-02
> >
> > Thank you for providing the additional experiments and clarifications, and my concerns have been addressed. After careful consideration, I will maintain my current score.

---

> > > ### Author Response · Authors · 2026-04-07
> > >
> > > We thank the reviewer for the continued engagement and for acknowledging that our previous response addressed your initial concerns. We truly value the time and effort you have dedicated to evaluating our work.
> > >
> > > To briefly summarize, based on your highly constructive feedback, we have made the following concrete improvements and clarifications to our manuscript:
> > > - **Data Curation Dependency**: We clarified that the reliance on external LLMs for data curation represents a strictly one-time, offline cost, which is a standard and well-justified investment for reasoning distillation.
> > > - **Preservation of Deep Reasoning**: We elaborated that CoSMo explicitly protects valid, complex reasoning chains by leaving intra-segment token length unconstrained, ensuring that essential reasoning depth is never suppressed.
> > > - **Strict Experimental Control**: We clarified that all baseline methods and CoSMo utilize the exact same GRPO algorithm, ensuring performance gains are solely attributed to our novel segment-level design.
> > > - **Evaluation Reliability**: We verified the high reliability of our selected LLM judge, supported by both recent comprehensive literature and our own manual human validation (achieving 95% consistency).
> > > - **Deeper Analyses and Generalization**: We incorporated a detailed analysis of algorithm failure modes and provided robust new experimental evidence demonstrating CoSMo's zero-shot generalization capabilities in Out-of-Distribution mathematical domains (GSM8K and MATH datasets).
> > >
> > > We hope these concrete improvements thoroughly resolve your concerns and might encourage you to consider raising your score. We remain fully available for any further discussion and would be happy to address any remaining subtleties you may have noticed.

---

### Official Review · Reviewer_bBT7 · 2026-03-13

**Soundness:** 2
**Presentation:** 3
**Significance:** 3
**Originality:** 3
**Overall Recommendation:** 3
**Confidence:** 3

**Summary:**

This paper introduces CoSMo, a framework designed to improve the reasoning efficiency of Large Reasoning Models (LRMs) in multi-hop Question Answering tasks. Instead of naively penalizing total token generation length, CoSMo explicitly decouples structural complexity (number of reasoning segments, N) from intra-segment token length. The framework consists of two stages: a Split-Merge algorithm to curate Supervised Fine-Tuning (SFT) data, and a reinforcement learning  phase utilizing a segment-level penalty. Experiments demonstrate that this approach effectively reduces token and segment consumption while preserving model accuracy.

**Compliance With Llm Reviewing Policy:**

Affirmed.

**Final Justification:**

The author’s response completely addressed my concerns, but considering that the paper still has some flaws, I will maintain my score.

**Key Questions For Authors:**

see weaknesses

**Limitations:**

yes

**Strengths And Weaknesses:**

Strengths:
- The article applies the classic split-merge algorithm to logical chain optimization and demonstrates, through case studies, how this method eliminates redundant verification steps while preventing logical leaps.
- The authors conducted experiments across a wide range of baseline models. The experimental validation goes beyond standard benchmarks, showing that as the intrinsic reasoning complexity (number of hops) increases, CoSMo maintains a stable accuracy rate.

Weaknesses:
- The entire framework, including the split-merge optimization and RL reward, requires prior knowledge of the optimal logical depth k. While k can be extracted in multi-hop QA datasets like HotpotQA where the number of hops is explicitly labeled, it remains undefined for tasks that do not specify reasoning complexity in advance. This dependency severely limits the framework's general applicability.

- As detailed in Appendix A.3, the split-merge optimization requires iteratively invoking a massive 72B model both as a judge and a generator to curate data for a much smaller 8B policy model. Is the computational cost too high?

---

> ### Author Rebuttal · Authors · 2026-03-30
>
> We thank the reviewer for the insightful, valuable, and positive comments. We address the concerns in detail as follows. We sincerely hope that our response could properly address your concerns. If so, we would deeply appreciate it if you could raise your score. If not, please let us know your further concerns, and we will continue actively responding to your comments and improving our submission.
> ### Weakness 1
> >The entire framework, including the split-merge optimization and RL reward, requires prior knowledge of the optimal logical depth k. While k can be extracted in multi-hop QA datasets like HotpotQA where the number of hops is explicitly labeled, it remains undefined for tasks that do not specify reasoning complexity in advance. This dependency severely limits the framework's general applicability.
>
> We appreciate the reviewer highlighting this valid limitation. We address the generalization of CoSMo from two perspectives: zero-shot generalization within QA tasks, and universal adaptation to Out-of-Distribution (OOD) non-QA domains.
>
> **Generalization across QA tasks:** We first want to clarify that CoSMo is primarily designed and optimized for complex QA tasks. Within this scope, while obtaining k* for arbitrary benchmarks is challenging, our empirical results demonstrate that the "low structural redundancy" trait learned on multi-hop QA generalizes remarkably well to OOD QA datasets like NQ and CRAG in a zero-shot manner, maintaining superior accuracy and efficiency. This indicates the model has successfully internalized a rigorous reasoning style by articulating necessary steps while rejecting redundancies. Thus, training on domains where k* is available acts as an effective catalyst to unlock a generalized, efficient reasoning paradigm.
>
> **Adaptation to general reasoning domains:** Furthermore, to universally apply our framework to distinct domains such as mathematics (e.g., GSM8k and MATH), we demonstrate that the algorithm can be easily modified to eliminate the reliance on k*. By simply skipping the parts of that require k* judgment, it iteratively applies merge and split operations directly on the reasoning chain based purely on the semantic judge's local verification. This process continues until the number of segments naturally converges, or until the maximum iteration limit is reached.
>
> To validate this adaptation, we conducted an experiment on the GSM8k training set (7.5k samples) using this k*-free iterative curation followed by SFT on the Llama-3.1-8B-Instruct backbone. For a fair comparison, we evaluated it against standard CoT and C3oT (a strong SFT-based pruning baseline) on the standard GSM8k test set under identical experimental settings. The results are summarized below:
>
> Method|Acc.|Tok.|Seg.
> -|:-:|:-:|:-:
> CoT|80.7|189|4.3
> C3oT|78.2|137|4.0
> CoSMo|84.6|153|3.7
>
> As shown in the table, even without the explicit global guidance of k*, CoSMo's self-guided split-merge process effectively eliminates structural redundancy (reducing segments to 3.7) while significantly boosting reasoning accuracy to 84.6%. This compellingly demonstrates that CoSMo is not strictly bound to k* annotations, and its core structural optimization philosophy can be broadly generalized to other complex reasoning benchmarks. We will include this extended discussion and experiment in the final version of the paper.
> ### Weakness 2
> >As detailed in Appendix A.3, the split-merge optimization requires iteratively invoking a massive 72B model both as a judge and a generator to curate data for a much smaller 8B policy model. Is the computational cost too high?
>
> We thank the reviewer for raising this practical concern. While iteratively invoking a 72B model incurs non-trivial computational overhead during data curation, this approach strictly aligns with the prevailing paradigm in reasoning distillation and CoT compression.
>
> A review of recent literature reveals that leveraging massive frontier models to curate high-quality reasoning trajectories for smaller policies is a standard and necessary practice. For instance, state-of-the-art methods such as C3oT, CtrlCoT, Extra-CoT, and MACC all heavily rely on the GPT-4 series for data generation and verification. Similarly, R1-Compress utilizes the Llama-3.1-70B-Instruct model.
>
> Crucially, this intensive computation is a one-time, offline cost incurred exclusively during the data preparation phase. This upfront overhead is well justified, as it directly enables a compact 8B model to deliver superior reasoning capabilities and structural conciseness during real-time inference.

---

> > ### Author Rebuttal · Reviewer_bBT7 · 2026-04-04
> >
> > The author’s response completely addressed my concerns, but considering that the paper still has some flaws, I will maintain my score.

---

> > > ### Author Response · Authors · 2026-04-07
> > >
> > > We thank the reviewer for the continued engagement and for acknowledging that our previous response addressed your initial concerns.
> > >
> > > Regarding your note that the paper still has some flaws, we would be grateful if you could specify them so we can address them immediately. As detailed in our rebuttal, we have made every effort to strengthen the submission, specifically by **detailing CoSMo's zero-shot generalization in QA tasks, providing new experimental results for its adaptation to OOD domains (GSM8K), and clarifying that the computational overhead is a strictly one-time offline cost, aligning perfectly with the prevailing paradigm in reasoning distillation and CoT compression.**
> > >
> > > We hope these concrete improvements thoroughly resolve your concerns and might encourage you to consider raising your score. We remain fully available for any further discussion.

---

### Official Review · Reviewer_qJTq · 2026-03-23

**Soundness:** 3
**Presentation:** 4
**Significance:** 3
**Originality:** 3
**Overall Recommendation:** 4
**Confidence:** 2

**Summary:**

This paper focuses on multi-hop reasoning datasets, motivated by the observation that the reasoning structure needs to better align with a problem's inherent structure in order to yield the best performance. Given this observation, the paper proposes curating datasets to follow the problem's inherent structure, as well as designing an RL reward function that encourages such structural alignment. The results support the effectiveness of this observation and the resulting training algorithm.

**Compliance With Llm Reviewing Policy:**

Affirmed.

**Key Questions For Authors:**

* The merge and split operations are applied sequentially from the first segment to the last. But could there be alternatives? Imagine that the latter segments are more reasonable candidates to be split or merged, yet the judge instead acts on the first few segments. How sensitive is the algorithm to this traversal order?
* The paper argues against token-level optimization, but this seems to conflate aggressive length penalties with milder regularization. After the split-merge algorithm correctly structures the reasoning, could each segment also be rewritten to be more concise? Similarly, could a weaker token-length regularization term complement the segment-level budget in RL, capturing the best of both worlds?
* The paper reports average segment counts, but not the deviation from the ground-truth hop count k* across all datasets. For HotpotQA and MuSiQue the alignment is shown, but what about HaluEval, NQ, and CRAG? Do these datasets even have k* annotations?

**Limitations:**

Yes.

**Strengths And Weaknesses:**

Strengths:
* This paper is well-motivated with clear preliminary experiments showing the benefits of aligning the number of segments (CoT steps) to the number of hops in the multi-hop QA dataset.
* The experiments across datasets have shown the effectiveness of this method: aligning the number of CoT steps helps improve the performance of reasoning.

Weaknesses:
* Though the reduction in the number of segments is very effective, the model still generates more tokens than some of the baseline methods compared.
* The entire framework — both the split-merge data curation and the RL reward — requires knowing k* at training time. This is available in multi-hop QA datasets but is not a common annotation in most reasoning benchmarks. How would the author generalize the algorithm to potentially other reasoning benchmark?

---

> ### Author Rebuttal · Authors · 2026-03-29
>
> We thank the reviewer for the insightful, valuable, and positive comments. We address the concerns in detail as follows. We sincerely hope that our response could properly address your concerns.
> ### Weakness 1
> >Though the reduction in the number of segments is very effective, the model still generates more tokens than some of the baseline methods compared.
>
> We thank the reviewer for this observation. While CoSMo does not yield the absolute minimum token count, this is a deliberate design choice. **By decoupling structural complexity from intra-segment elaboration, we avoid blindly compressing individual steps, which often forces the omission of crucial details and restricts reasoning enhancement**. Despite preserving this necessary depth, CoSMo reduces overall token consumption by 29.5% against standard CoT and achieves the highest accuracy by eliminating structural redundancy. This establishes a vastly superior trade-off between inference efficiency and complex reasoning, which we will clarify in the final manuscript.
> ### Weakness 2
> >How would the author generalize the algorithm to potentially other reasoning benchmark?
>
> We appreciate the reviewer highlighting this valid limitation. Regarding how the algorithm generalizes to other reasoning benchmarks, please refer to our detailed response to **Weakness 1 of Reviewer bBT7**.
> ### Question 1
> >The merge and split operations are applied sequentially from the first segment to the last. But could there be alternatives? Imagine that the latter segments are more reasonable candidates to be split or merged, yet the judge instead acts on the first few segments. How sensitive is the algorithm to this traversal order?
>
> We appreciate this insightful question. Algorithmically, operations are strictly governed by our semantic gatekeeper ($\mathcal{M}_{\text{judge}}$) rather than segment position, ensuring all valid candidates are evaluated regardless of traversal direction.
>
> Empirically, an ablation study on HotpotQA comparing forward and backward traversal shows:
> Traversal Order|Acc.|Tok.|Seg.|
> -|:-:|:-:|:-:
> Forward|84.7|98|2.1
> Backward|83.2|93|2.1
>
> **Both orders achieve an identical segment count, proving that structural compression is robust to direction. While backward traversal yields slightly lower accuracy, performance remains highly competitive**. This confirms CoSMo's efficacy is largely insensitive to traversal order, and we will include this in the appendix.
> ### Question 2
> >The paper argues against token-level optimization, but this seems to conflate aggressive length penalties with milder regularization. After the split-merge algorithm correctly structures the reasoning, could each segment also be rewritten to be more concise? Similarly, could a weaker token-length regularization term complement the segment-level budget in RL, capturing the best of both worlds?
>
> We highly appreciate this constructive suggestion. While our current design strictly decouples token-level optimization to avoid suppressing deep reasoning, we acknowledge this unavoidably introduces verbosity within individual segments.
>
> As astutely pointed out, **combining a milder token-length regularization term with the segment-level budget by effectively assigning independent weights to intra-segment and inter-segment regularizations is an excellent approach**. This allows the model to secure the correct logical topology first and subsequently refine the conciseness of each step. We have incorporated this valuable insight into the Limitations and Future Work section of the revised manuscript.
> ### Question 3
> >The paper reports average segment counts, but not the deviation from the ground-truth hop count k* across all datasets. For HotpotQA and MuSiQue the alignment is shown, but what about HaluEval, NQ, and CRAG? Do these datasets even have k* annotations?
>
> We clarify that HaluEval, NQ, and CRAG lack official k* annotations, making exact deviation calculations infeasible. However, based on task characteristics, we estimate the requisite k* to be 2–3 for HaluEval and 2–4 for NQ and CRAG. **Among all evaluated methods, CoSMo generates segment counts that are significantly closest to these expected ranges, demonstrating its ability to adaptively align structural complexity with task difficulty even without explicit k\* supervision**.

---

### Decision · Program_Chairs · 2026-04-30

**Decision:**

Accept (regular)

**Comment:**

This paper aims to design a framework to refine reasoning chains via a split-merging algorithm. All reviewers admit the contributions of this paper with simple idea to decouple structure from reasoning and provide effective experiments. During the rebuttal stage, reviewers raises these concerns:

- Reviewer qJTq: 1. Efficiency as model still generate more tokens; 2.  Generalization (Dependency of K).
- Reviewer bBT7: 1. Dependency of K; 2. Efficiency.
- Reviewer 5AEd: 1. Efficiency (LLM for data generation); 2. Assume target are optimal; 3. Baseline; 4. Evaluation with high variance; 5. generalization.
- Reviewer AdJP: 1. Baseline / Benchmark; 2. Discussion on visual reasoning.

After the rebuttal stage, most of reviewers' concerns have been addressed while Reviewer bBT7 point this paper still have few flaws but not specific it. After reviewing the all reviewers' response, the main concerns of this paper is 1) efficiency, but data generation is one-time cost; 2) Rely on K. For efficiency, I suggest authors should also report the cost of data generation in the paper. For k, as multi-hop QA tasks could have specific K, it can be ignored but still need to be considered in other tasks. Authors claim that for some reasoning tasks, can directly skip the parts of that require k* judgment. In total, I suggest authors should highlight their task scope for some specific reasonings. As the define of  long reasoning tasks is wide, which will bring some unclear understanding. Overall, I recommend this paper as weak accept as it have some minor issues but not affect the contribution of this paper.